# Carotenoids Diet: Digestion, Gut Microbiota Modulation, and Inflammatory Diseases

**DOI:** 10.3390/nu15102265

**Published:** 2023-05-10

**Authors:** Helena R. Rocha, Marta C. Coelho, Ana M. Gomes, Manuela E. Pintado

**Affiliations:** CBQF–Centro de Biotecnologia e Química Fina–Laboratório Associado, Escola Superior de Biotecnologia, Universidade Católica Portuguesa, Rua Diogo Botelho 1327, 4169-005 Porto, Portugal; s-mhrocha@ucp.pt (H.R.R.); amgomes@ucp.pt (A.M.G.)

**Keywords:** carotenoids, gastrointestinal tract, intestinal microbiota, metabolites, absorption

## Abstract

Several epidemiologic studies have found that consuming fruits and vegetables lowers the risk of getting a variety of chronic illnesses, including several types of cancers, cardiovascular diseases (CVDs), and bowel diseases. Although there is still debate over the bioactive components, various secondary plant metabolites have been linked to these positive health benefits. Many of these features have recently been connected to carotenoids and their metabolites’ effects on intracellular signalling cascades, which influence gene expression and protein translation. Carotenoids are the most prevalent lipid-soluble phytochemicals in the human diet, are found in micromolar amounts in human serum, and are very susceptible to multiple oxidation and isomerisation reactions. The gastrointestinal delivery system, digestion processes, stability, and functionality of carotenoids, as well as their impact on the gut microbiota and how carotenoids may be effective modulators of oxidative stress and inflammatory pathways, are still lacking research advances. Although several pathways involved in carotenoids’ bioactivity have been identified, future studies should focus on the carotenoids’ relationships, related metabolites, and their effects on transcription factors and metabolism.

## 1. Introduction

Carotenoids are natural pigments from the tetraterpenes family, characterized by a central chain with 40 atoms of carbon and alternating single and double bonds and various cyclic or acyclic end groups, depending on the carotenoid [1]. In terms of physicochemical properties, carotenoids are colourful lipophilic compounds [1,2], responsible for the variety of colours present in several autotrophs such as microalgae, bacteria, fungi, and plants [1,3].

Humans and animals cannot synthesize carotenoids by themselves, they can be found in their tissues due to the absorption and deposition of the carotenoids ingested in dietary food [2,4,5].

Carotenoids are natural organic pigmented compounds with structural variations, with more than 750 carotenoids being known, but only 40 of them are present in the human diet and 20 in human blood and tissues [1,6,7,8]. The 40 carotenoids present in a usual human diet [1] can be found in coloured fruits and vegetables, such as tomatoes, carrots, and spinach [9].

In terms of chemical constituents, these natural pigments can be divided into two categories: carotenes and xanthophylls [10]. If they are pure hydrocarbons, they can be classified as carotenes such as alpha(α)-carotene, beta(β)-carotene, and lycopene [8]. Xanthophylls are carotenoids with oxygenated derivatives on their terminal rings [8], with complex xanthophylls containing oxygen substituents, aldehyde groups, epoxide groups and oxo/keto groups [11]. Zeaxanthin, lutein, canthaxanthin, violaxanthin and neoxanthin are examples of complex xanthophylls. 

Carotenes absorb light energy from chlorophyll and energy from singlet oxygen formed in photosynthesis, being responsible for transmitting this light and protecting the plant tissues [5]. Xanthophylls, synthesized within the plastids, work as accessory pigments, capturing the wavelengths of sunlight that chlorophyll cannot absorb [5].

Regarding functional properties, carotenoids can be classified as primary and secondary carotenoids, with the photosynthetic ones included in the primary group and playing an important role in photosynthesis [2,12].

These natural organic pigmented compounds, in terms of physicochemical properties, are associated with membrane lipid bilayers and cytosolic lipid droplets, which can affect some properties associated with the permeability and fluidity of the membrane [9].

The principal properties of carotenoids mentioned before are illustrated below, in Figure 1.

The regular consumption of fruits and vegetables is widely recommended due to their multiple health benefits such as the lower incidence of chronic diseases [13] such as cardiovascular diseases (CVDs), several types of cancers [14], and bowel diseases. Chron’s disease and ulcerative colitis are two chronic inflammatory bowel diseases (IBDs) characterized by recurring episodes of inflammation in the gastrointestinal tract (GIT) [15] that cause damage to its tissues [16].

Several studies have attributed to bioactive compounds present in the diet [14], in particular carotenoids [13,17], the responsibility for beneficial health effects in various pathologies, namely IBDs. This can be explained by their several important biological functions such as antioxidant activity [2,18], meaning that these pigments can inhibit or downregulate the unstable compounds produced by the body [2,18] in various pathologies and during oxidative stress caused by reactive oxygen species (ROS) [1,19]. In addition to this, carotenoids have other important functions such as antibacterial, immunological, and anti-inflammatory activity, and beneficial effects on the treatment of diabetes, and in infectious, eye, and neurological diseases [2,18].

Some of the more important biological functions related to human health of the three most known carotenoids are presented in Table 1.

These antioxidant phytochemicals are also important dietary sources of vitamin A and protect cells and tissues from oxidative damage, interacting with other antioxidants [11,25]. So far, only 50 carotenoids are known to have provitamin A activity [12], with α-carotene, β-carotene, gamma(γ)-carotene, and β-cryptoxanthin being the most important precursors of vitamin A in humans [12,20]. 

Vitamin A is important for proper visual, immune, and gastrointestinal functions, growth, and embryonal development [20]. Humans cannot synthesize vitamin A de novo, obtaining adequate amounts through dietary food, such as from orange and yellow vegetables and in vegetables with dark green leaves [20]. 

The recent discoveries about the health promotion properties of carotenoids have aroused interest in applying these natural pigments in diversified areas [2]. These natural pigments have several applications such as in feed, and in the food, nutraceutical, and pharmacology industries [5]. Carotenoids can be applied as colourants in food, beverages, and cosmetics, as food supplements, as feed additives, and as supplements [26].

In nature, the bioavailability of carotenoids is reduced [1,20] without processing or any type of treatment, leading to an accumulation in the colon [27], which is colonized by a large number of microorganisms [28] that play important roles in digestion and metabolism [29], as well as in maintaining normal gut physiology and health [30]. 

Diet is one of the most important regulators of the intestinal microbiota [31], but there is a lack of information about the relationship between carotenoids and the intestinal microbiota [32]. In addition to that, these natural pigments are hydrophobic molecules, which makes their solubility in water low and, when exposed to light, heat, oxygen, or acids, are very susceptible to multiple oxidation and isomerisation reactions [5]. Therefore, the polarity of carotenoids can change due to the polar functional groups attached to the main chain and some products with harmful or unknown effects can also be formed [1], which can affect the carotenoids’ bioaccessibility, bioavailability, and absorption.

In this sense, this review aims to present an overview of the gastrointestinal delivery system for carotenoids, the processes occurring during digestion, from mastication to absorption, and the impact of the gut microbiota and its metabolites on the stability and functionality of carotenoids, and also their ability to modulate inflammatory and oxidative stress pathways.

## 2. Overview of the Publications

### 2.1. Methodology of Research

The research articles about carotenoids and intestinal microbiota were searched on Science Direct and on PubMed, using the keywords “carotenoids”, “gut microbiota”, and “interaction”. Therefore, all publications available in these two databases which contained the words mentioned before as author-specified keywords in the title or abstract were considered.

Later, the abstracts of all of the articles were analysed and divided into two categories: carotenoids in human health and the interaction of carotenoids with the gut microbiota. There are several articles that mention that phytochemicals/dietary lipids may influence the composition of the intestinal microbiota, the digestion process, and the occurrence or prevention of some diseases, among others. Although carotenoids belong to those categories, the detailed process is focused on other phytochemicals/dietary lipids. For carotenoids, the process is only a supposition, according to the chemical and structural similarities with other dietary lipids, highlighting the lack of information about these natural pigments. For this reason, articles where the role or the process that carotenoids perform is not the main focus were excluded. 

### 2.2. Results

According to our search through the Science Direct database, only four relevant research articles were published (2018: 1; 2022: 3). The oldest article was published in 2018, which reveals that the interaction between carotenoids and the intestinal microbiota is a recent research theme.

The search performed on PubMed showed that 18 articles containing the keywords selected by us were published (2017: 2; 2018: 3; 2019: 2; 2020: 6; 2021: 0; 2022: 6). Although more articles were found through *PubMed* than through Science Direct, it corroborates the fact that this topic is a very recent research target and that more research needs to be conducted concerning the interaction between carotenoids and the intestinal microbiota. The articles published were categorized into carotenoids in human health (5 articles) and the interaction of carotenoids with the gut microbiome (7 articles).

The number of published articles related to carotenoids and their interaction with the intestinal microbiota, according to its year of publication, is represented in Figure 2.

Initially, the studies were more related to the role that carotenoids play in human health such as the antioxidant and anti-inflammatory activity and the beneficial effects that they have on the treatment of some cancers and CVDs. It is believed that due to these bioactive properties, the interest in carotenoids has increased exponentially, which is corroborated by the increase in articles about the interaction of carotenoids with the gut microbiome in more recent years. 

However, it still lacks information related to the impact of digestion and interaction between the carotenoids and the gut microbiome on the stability and functionality of carotenoids. 

## 3. Bioaccessibility and Bioavailability of Carotenoids

The bioaccessibility of a carotenoid is defined as the maximum quantity of a carotenoid released from the food matrix that is available to be absorbed in the epithelial cells of the intestine [33]. The fraction of an ingested compound that enters the bloodstream and performs its physiological functions is the definition of the bioavailability of a carotenoid [5,34]. 

In nature, the bioavailability of these natural pigments is reduced, since there is a resistance to digestion and degradation from the protein complexes of carotenoids and the cell walls of plants to achieve adequate release from the matrix [1,20]. In the case of β-carotene, the activity and conversion to vitamin A are high. However, the absorption from plant sources is approximately 65%, with the recommended daily intake of 2–4 mg per day not being achieved [20,35]. 

The carotenoids’ bioavailability and consequently absorption have some limitations due to factors such as the dietary sources, food composition, cooking temperature, season, the breakup of the food matrix, presence of lipids, dosage, and presence of other soluble compounds/carotenoids [18,36]. These factors can lead to the release of carotenoids from the food matrix, improving its bioavailability or transforming the carotenoids into isomers that are better absorbed by the organism [36].

The release from the food matrix depends on the state of the carotenoid, as natural pigments immersed entirely in lipid droplets are more easily released than ones in the microcrystalline form [37]. This explains the low availability of lycopene in tomatoes and β-carotene in carrots [37]. 

The dietary composition also has a significant effect on the bioavailability of carotenoids [36]. Carotenoids are lipophilic compounds and for this reason their bioavailability increases when they are consumed allied with a fat source [38], but decreases when they are consumed with dietary fibre such as pectin [39]. 

Food thermal processes such as cooking, boiling, and heating disrupt the cellular membrane, allowing the release of carotenoids from the matrix [1,20,21], and although this leads to a decrease in the carotenoid content, it raises their bioavailability and absorption when compared with uncooked food [1,8,20]. For example, in cooked tomatoes, the lycopene availability is higher than in raw tomatoes, and the more prolonged the heat treatment, the lower the carotenoid content is [21].

The principal factors affecting the carotenoids’ bioavailability, enhancing (left) or decreasing (right) it, are represented below, in Figure 3.

Therefore, different extraction technologies are required to increase carotenoids’ solubility and bioavailability [40]. The traditional methodology uses organic solvents such as hexane and acetone to extract carotenoids from food matrices, because of their hydrophobicity [40]. However, the toxicity of these organic solvents to human health, imposes the use of food-grade solvents to purify these carotenoids and use them in the food industry [40].

In the last few years, some alternative methods to recover carotenoids have been presented, such as super-critical fluid extraction (SFE), high hydrostatic pressure (HHP), and Ohmic heating (OH).

SFE is an extraction technique that reduces the toxic solvents used during the process and can generate a solvent-free extract at moderately high selectivity and yield temperatures [11]. Although it is a non-inflammable and non-toxic method, its non-polar nature demands the use of a stabiliser and a cosolvent, and carotenoid degradation and/or isomerization can occur [40,41]. This technique is advantageous insofar as the process is both environmentally benign and energy efficient and the sustainable solvent is simple to obtain. However, it presents some limitations, since it is an expensive method and the polar extracts are insoluble in the CO_2_ mobile phase [42,43,44].

HHP is a simpler and more efficient technique than conventional extraction methods, that contributes to improve the bioaccessibility of bioactive compounds [45]. HHP is advantageous since it is a completely solvent-free procedure that uses tomato leaf waste at a high CO_2_ pressure (180 bar), and at room temperature to obtain phylloquinone [41]. However, once again it is limited by its high cost, and by the necessary improvements in the associated recovery process [46].

More recently, Coelho et al. [47] proposed OH, which consists of the use of an electric current that passes across a conductor matrix (e.g., food) to generate heat from the electrical resistance of the matrix. This methodology is more advantageous than the ones mentioned before since it allows the extraction of bioactive compounds such as carotenoids and polyphenols from their matrices only using ethanol:water as a solvent [45,47], and the application of a low temperature prevents thermal losses [11]. 

The authors [47] showed that this method can replace traditional methods since it is selective, enabling bioactive compounds to be extracted without organic solvents. OH has some limitations given the impossibility of extracting some bioactive compounds that remain bound to dietary fibres and the lack of information about the potential antioxidant properties of these bioactive compounds, as well as how they are affected by the GIT during digestion [48].

The main advantages and limitations of OH as an alternative method to recover carotenoids without organic solvents are represented in Figure 4.

## 4. Carotenoid Absorption Mechanism

The carotenoids’ pathway along the GIT starts in the mouth, where they are liberated from the food matrices and, passing through the stomach and intestine, become susceptible to modifications such as solubilization by the intestinal fluids [48,49]. Then, the bioactive compounds in the intestine suffer selection through permeation, becoming available for bloodstream absorption [48], and the non-bioaccessible ones are directly used by the gut microbiota [48,49].

The carotenoid absorption mechanism can be divided into release from the food matrix, transfer to the oil phase, formation of mixed micelles, and absorption, as represented below, in Figure 5.

### 4.1. Release from the Food Matrix

The carotenoid absorption mechanism starts with mastication, the physical disruption that leads to the release of carotenoids from the food matrix [20]. This step is the first limiting factor affecting bioavailability since the physical form of carotenoids conditions their release during digestion [20,50]. In the case of β-carotene, for example, it can be within the food in liquid crystalline form, such as in mango and papaya, or in solid crystalline form, as in carrot and tomato [20,50], with the bioavailability of this carotenoid in food being higher in the liquid crystalline form [20,50].

### 4.2. Transfer to the Oil Phase

The second step consists of the dissolution of carotenoids into the gastric emulsion. The first limiting factor is the digestion from the food matrix: if it is not complete, carotenoids will not have direct contact with the oil and, consequently, will not be transferred to the oily phase [20].

The incorporation of carotenoids into the gastric emulsion also faces several limiting factors, such as soluble proteins, the surface charge of the gastric emulsion, the oil, and the amount of the carotenoid present [20].

In the case of β-carotene, the incorporation of this carotenoid into the gastric emulsion is inhibited by soluble proteins that affect the interfacial characteristics of the digesta. Proteins (e.g., caseins) have been suggested to help in the bioaccessibility of liposoluble food elements in a variety of ways. Proteins may stabilize oil-in-water (o/w) emulsions in the GI tract after adsorption to lipid droplet surfaces. This is due to the fact that proteins can be highly surface-active molecules, and the formed particles tend to be highly negatively charged, preventing lipid droplet aggregation. However, Qiu et al. [51] found that gliadin reduced enzymatic lipid degradation, most likely by preventing digestive enzymes from adsorbing to droplet surfaces or directly binding to enzymes, implying that proteins may have a negative influence on the micellization process. Whey protein isolate (WPI) inhibits lipid oxidation and facilitates the formation of smaller lipid droplets, increasing β-carotene bioaccessibility. WPI has both hydrophobic and hydrophilic groups, and its conformation affects its properties at the oil/water interface. This study investigated the effect of WPI on the bioaccessibility of pure carotene under different digestive conditions. Micellization of co-digested β-carotene was also measured under insufficient digestion parameters [52].

However, the concentration of soluble proteins decreases and the transfer of β-carotene to oil increases if the pH decreases [20]. β-carotene’s solubilization increases when the gastric emulsion’s surface charge decreases, since it allows a higher adherence of oil to the carotene-containing matrix [20]. In addition to that, the oil, and also the amount of the carotenoid present, affect the solubility of β-carotene in the oily phase, which determines the extension of the carotenoid transfer to the digesta [20].

### 4.3. Micelle Formation

During the passage through the small intestine, the release of bile salts occurs that promotes the formation of mixed micelles [20]. These micelles are the result of the action of bile salts as surfactants that reduce the size of the gastric emulsion, composed of free fatty acids, monoglycerides, phospholipids, and the carotenoid [53], to micelles with an 80 Å diameter, approximately [53]. The micelles have an amphiphilic structure that allows the lipophilic nutrients to be remain soluble in the aqueous digesta [36].

The carotenoids’ absorption only occurs if they are in mixed micelles, since the factors that affect the micelle formation also affect the bioavailability of carotenoids in the digestion process [20]. Dietary fat is a factor that influences the formation of micelles, since lipids are necessary to stimulate the release of bile and for the incorporation of the gastric emulsion into micelles [20]. However, Roodenburg et al. [54] indicated that increasing dietary fat is only beneficial to the formation of micelles until an optimal threshold. In addition to this, the fat type affects the micelle formation, since the longer the fatty acyl chain, the more extensive the micelle formation and the bioavailability of the carotenoids [20].

In addition to lipids, fibres such as alginate, guar, and pectin are also limiting factors, as in the presence of carotenoids they inhibit the formation of micelles and decrease the bioavailability of carotenoids [20].

### 4.4. Absorption

The final step of the carotenoids’ absorption starts when the micelles containing the carotenoid come into contact with the apical side of the enterocytes [20], then enter the enterocytes, are incorporated into chylomicrons with other dietary lipids, and are transported across the basolateral membrane [55]. Then, the carotenoid enters the lymphatic system and is released into the circulation, being distributed throughout the body [36,55].

Although it was believed that the absorption of carotenoids occured in the same way as dietary lipids, through passive diffusion, it has been discovered that the absorption of carotenoids can be facilitated by transporters present in the membrane, such as the scavenger receptor class B type 1 (SR-BI), the cluster determinant 36 (CD36), and NPC1-like transporter 1 (NPC1L1) [9,55]. SR-B1 is a class B receptor found in different tissues, particularly in the intestine, and is involved in the cellular uptake of a wide range of lipid molecules (e.g., cholesterol and liposoluble vitamins) [56] and of the non-provitamin A carotenoids lutein [57] and lycopene [58]. CD36 is present in various tissues, namely in the intestine, and has ligands for carotenoids, long-chain fatty acids, and lipoproteins [56]. CD36 and SR-B1 are glycosylated transmembrane proteins with a large extracellular domain [59]; it has been predicted that CD36 has a large cavity traversing its entire length that allows lipid transfer from extracellular to cellular compartments [60,61]. NPC1L1 is a major sterol transporter in the intestine [62,63] and is involved in the uptake of carotenoids such as α-carotene, β-carotene, β-cryptoxanthin, and lutein [64,65]. Although some studies proved that these proteins have facilitated carotenoid absorption [59], the mechanisms behind these effects are still unknown [56].

In addition, absorption can be affected by the individual’s genetic susceptibility, the dose ingested [1], and by viscosity, since it inhibits the formation of micelles and consequently decreases the amount of carotenoid available in a form capable of absorption [20]. 

## 5. Intestinal Microbiota

The intestinal microbiota consists of a complex community of microorganisms [30], including bacteria, viruses, and some eukaryotes that live in the digestive tracts of humans and animals [66]. The number of bacterial cells in the human intestinal microbiota is approximately 10^14^, which is 10 times higher than the number of human cells [67]. 

The intestinal microbiota composition is different along the GIT [30]. The stomach and small digestive tract are colonized by only a few species of bacteria, and in the colon are present approximately 10^12^ bacterial cells/g of gut content [30]. Almost 99% of the bacteria that colonize the intestine are anaerobes, but in the connection between the small intestine and the colon (cecum), a high density of aerobic microorganisms can be found [30]. 

In Figure 6 are presented the main microbial species and their respective density present in the different parts of the GIT.

The relationship between the intestine and its microorganisms is mutualistic, since the host intestine supplies the bacteria with the conditions for their survival and reproduction and the microbiota has important functions such as digestion, nutrient processing, protection against pathogens, production of different antimicrobial substances [30], production of micronutrients such as vitamins, immune cell growth and response [68], and the control of epithelial cell proliferation and differentiation [69].

The intestinal microbiota has a spatial limitation, which consists of its enclosure within the gastrointestinal lumen, allowing the gut bacteria to translocate and generate a local or/and systemic inflammation [70,71]. To overcome this limitation, intestinal microbiota release a large number of different metabolites [71], including bile acids, vitamins, amino acids such as tryptophan, and short-chain fatty acids (SCFAs) [72]. These metabolites will have extensive effects on a host’s organs near or far from the gastrointestinal lumen, such as regulating local and systemic immune response, nutrient absorption, host metabolism, and gut microbiota composition to maintain health or develop diseases [71].

Nonetheless, the composition and function of the intestinal microbiota can be affected [30] by individual intrinsic factors such as age, ethnicity, and genetic markers, or by environmental factors such as geographic area, lifestyle, diet, and drugs [73,74]. These factors, particularly diet, can lead to useful or harmful modifications in the production of the metabolites that could change the composition of the microbiota, increasing or decreasing some species present [30,75]. 

## 6. Intestinal Microbiota Metabolites

Dietary compounds with low bioavailability or that cannot be absorbed directly [76], pass the small intestine and enter the colon, where they will interact with live gut bacteria [31]. The colon is the ideal place for the interactions between the intestinal microbiota and the dietary compounds since, in addition to a high level and diversity of the microorganisms present, it supplies the suitable pH and time for direct contact between microbes and food [67]. Once there, these compounds can induce functional and compositional modifications of the microbiota or can be transformed into new compounds [31].

The intestinal microbiota contains millions of microbial genes [77] that enable the production of a large number of enzymes, which can ferment the dietary compounds that are not digested by human enzymes, such as fibre or primary bile acids [31]. In consequence, the intestinal microbiota are capable of synthesising and releasing a variety of different metabolites that can be produced directly from dietary compounds, produced by hosts and transformed chemically, or synthesized de novo by gut microbiota [78].

These plant-derived compounds must be absorbed, transferred to the circulating system, delivered to the site of action in the body, and metabolically converted to the vitamin active form to be biologically effective [79]. The carotenoid cleavage products (e.g., apocarotenoids) [80] are generated through the action of specific enzymes, such as the carotenoid cleavage dioxygenase (CCD) in plants [81] and the β,β-carotene-15,15′-oxygenase (BCO) in vertebrates [79]. However, the BCO enzymatic activity in the gut still needs to be clarified [79].

After biological activation, these metabolites can promote a wide range of activities in the host, such as regulating the composition, function, intestinal barrier and motility of the intestinal microbiota, modulating host metabolism, and influencing nutrient absorption, among others [71]. On the other hand, these metabolites, dependent on their chemical nature, can play important roles in the development and progression of diseases such as cancer, hypertension, Parkinson’s, and non-alcoholic fatty liver diseases [82,83,84,85]. 

In addition to that, dietary compounds can also indirectly interact with the intestinal microbiota through the modulation of gastrointestinal transit time, pH, and the synthesis and release of antimicrobial peptides and secretory immunoglobulins [86].

The typical metabolites generated by the intestinal microbiota and their respective functions and associated diseases are presented in Table 2. 

## 7. Interaction between Carotenoids and the Intestinal Microbiota 

The interaction between carotenoids and the intestinal microbiota is a topic that still lacks associated information and clear evidence. However, some studies have indicated that the intestinal microbiota may be the main factor behind the effectiveness of carotenoids’ action [120].

Jalal et al. [121] showed that the excessive growth of Proteobacteria, harmful bacteria, led to the damage of the mucosal epithelial cells and an increase in the permeability of the intestine, which provoked a decrease in the absorption of carotenoids. Another study, that used colonic fecal samples, showed that new compounds were generated during the fermentation of carotenoids by the intestinal microbiota, indicating that carotenoids were metabolized [122]. Although the absorption of carotenoids can be different depending on the individual [123], the studies mentioned before indicated that the composition of the intestinal microbiota has an important influence on the absorption and metabolism of carotenoids [120]. 

Other studies revealed that the composition of the intestinal microbiota can be regulated through dietary carotenoid supplementation such as lycopene, which inhibits the reproduction of Proteobacteria and promotes the growth of Bifidobacterium and Lactobacillus, maintaining the harmony of intestinal immunity and mitigating the symptoms caused by anxiety and dextran sulfate sodium-induced colitis and depression [124]. Astaxanthin has been associated with a relief of inflammation and a decrease in lipid accumulation, through a decrease in Bacteroidetes and Proteobacteria abundance and an increase in the population density of Verrucomicrobiota and Akkermansia sp. [125]. 

Supplementation with β-carotene also increased the abundance of Bacteroidetes and Proteobacteria and decreased the abundance of harmful bacteria such as Dialister and Enterobacter, which corroborates the positive effects of this carotenoid in intestinal health [120]. In addition, a dose of administered β-carotene also influenced the composition of the intestinal microbiota, since low and medium doses increased the abundance of *Bifidobacterium* and *Collinsella* strains and high doses increased the abundance of *Lactobacillus* strains [120].

In addition to this, the results obtained by Dai et al. [27] suggested that xanthophylls such as lutein and zeaxanthin have a higher impact on the modification of intestinal microbiota composition than carotenes. This demonstrates that carotenoids are structurally distinct and can affect differently the composition of the intestinal microbiota [27]. 

Therefore, the results obtained from these studies indicate that carotenoids and the intestinal microbiota have a structure–activity relationship and the latter can be a potential target for carotenoids’ utilization [27]. However, a comprehensive understanding of the direct interaction between carotenoids and the intestinal microbiota and their relationship is still lacking [126].

## 8. Carotenoid Metabolites from Microbiota and Activation/Deactivation of Gene Potentiation in Bowel Diseases

Carotenoids have been associated with various health benefits, mainly due to their anti-inflammatory and antioxidant properties that provide protection against lipid peroxidation and damage caused by ROS [127]. 

In addition to carotenoids’ scavenging function, it is believed that these natural pigments can also act indirectly [13]. This indirect pathway may include interactions with cellular signalling cascades, such as nuclear factor κB (NF-κB), mitogen-activated protein kinase (MAPK), or nuclear factor erythroid 2–related factor 2 (Nrf2) [51,52]. Some studies showed that carotenoids can be key players in NF-κB regulation, since they contain electrophilic groups that can interact with the cysteine residues of IκB kinase (IKK) and NF-κB subunits (p65), inactivating the NF-κB pathway and consequently decreasing the transcription of pro-inflammatory cytokine genes (e.g., TNF-α) [128,129]. In a recent study, Li et al. [130] showed that astaxanthin was able to protect retinal epithelial cells from H_2_O_2_-induced oxidative stress by inducing nuclear localization of Nrf2 and reducing intracellular ROS. However, in other studies, the Nrf2 translocation has been inhibited under different concentrations [13]. The effect of carotenoids in the MAPK pathway has been studied only in a few investigations and the results are very contradictory [13]. 

Despite the scarce number of investigations and the conflicting results obtained, there is also a major research gap related to carotenoid metabolism along the GIT and interactions with the gut microbiota [131]. Carotenoids are unstable molecules and are very susceptible to undergo various modifications such as hydrogenation, dehydrogenation, double-bond migration, chain shortening or extension, rearrangement, isomerization, oxidation, or combinations of these processes under different conditions [132]. 

Some studies showed that some unknown metabolites were produced during the carotenoids’ pathway through the GIT. These metabolites may include apocarotenoids, that have a shorter chain length and oxygen modification, which increases their aqueous solubility and electrophilicity, and consequently, improves the target of some transcription factors such as NF-κB, giving these metabolites some biological effects [127,128]. Furthermore, it was reported that certain microbes produce carotenoids in the colon, demonstrating a prebiotic-like effect that results in bacterial shifts with health-associated properties [127]. 

Since it has been proved that carotenoids are important for human health, it would be important to understand the mechanisms used by carotenoids to become available for absorption in the host colon, how they are utilized by microbes, and how carotenoids and their metabolites are processed to bring so many human health benefits [127]. This information will provide guidance to develop strategies for cell function manipulation through diet/nutraceuticals, impacting positively human health [132].

## 9. Conclusions

Carotenoids are natural pigments with important bioactive properties that promote health, becoming more studied and used over the last few years. Carotenoids’ bioavailability in nature is low and little is known about their pathway through the GIT and consequent processes occurring during digestion, as well as the role of the intestinal microbiota and their metabolites on the metabolism and absorption of carotenoids. 

Traditional methodologies such as SFE and HHP are used to increase carotenoids’ solubility and bioavailability, but the organic solvents used are toxic to human health, necessitating the purification of carotenoids that are to be used in the food industry. More recently, OH has been proposed as a more advantageous methodology that can replace the traditional methods since it is selective, enabling bioactive compounds to be extracted without organic solvents.

This review set out to give an overview of the carotenoids’ absorption mechanisms, mentioning the variables that can affect the stability and functionality of carotenoids. The four steps of such a mechanism were explained and the limiting factors that affect the bioavailability of these natural pigments in each phase were indicated. 

In addition, it was highlighted that the intestinal microbiota can have an important influence on the absorption and metabolism of carotenoids. The change in the composition of the intestinal microbiota can enhance or inhibit the reproduction of some microbial species that can have defensive or damaging effects. The physicochemical structure of these natural pigments, the co-consumption with other compounds, the host variables, and the presence and type of food matrix are examples of other factors that can play important roles in carotenoid efficiency.

The intestinal microbiota can synthesise and release a variety of different metabolites that can be produced directly from dietary compounds (e.g., cellulose), produced by hosts and transformed chemically by gut bacteria (e.g., bile acids), or synthesized de novo by gut microbiota (e.g., ATP). These metabolites are absorbed and transferred into the circulating system and can regulate the composition and function of the host’s intestinal microbiota, as well as play important roles in the development and progression of some pathologies.

However, just a few studies have been performed to understand the metabolism and absorption of these bioactive compounds along the GIT and further information and details about the mechanisms they use, as well as their metabolites, that contribute to human health benefits are still lacking. For this reason, more studies are required, since carotenoids have many important biological functions in the human organism, including the prevention of some of the most fatal diseases worldwide. 

## Figures and Tables

**Figure 1 nutrients-15-02265-f001:**
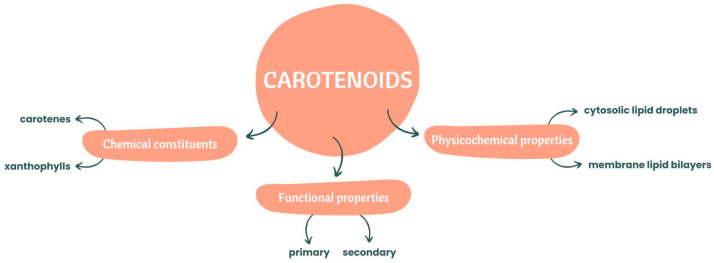
Chemical constituents and functional and physicochemical properties of carotenoids.

**Figure 2 nutrients-15-02265-f002:**
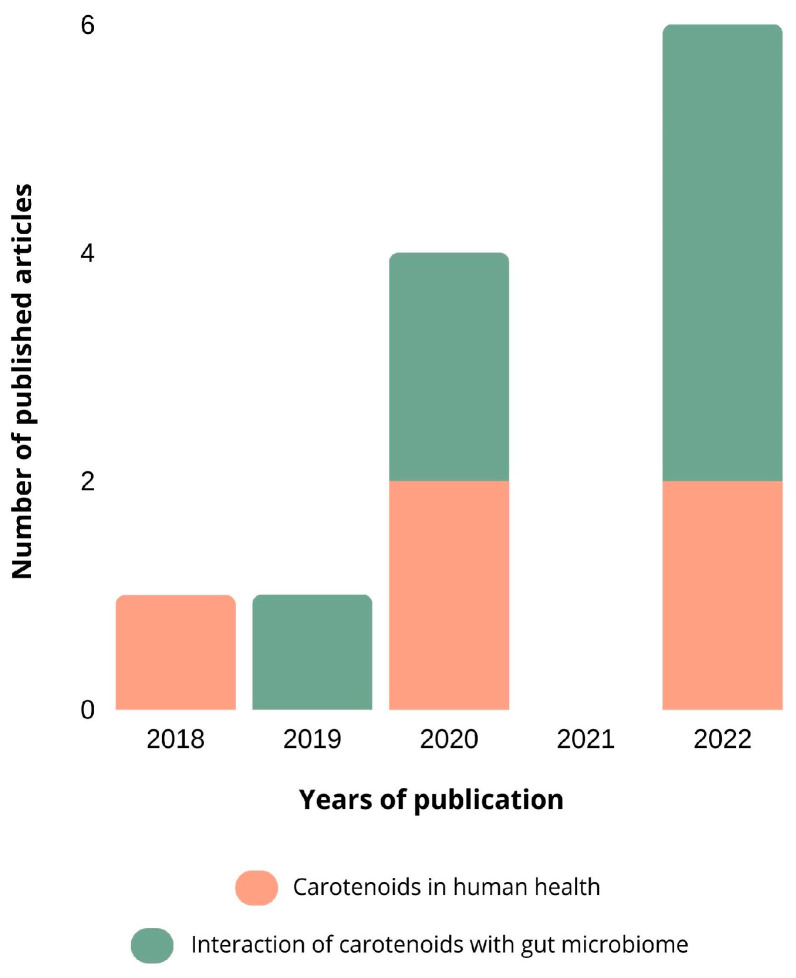
The number of published articles related to carotenoids and their interaction with the intestinal microbiota. The databases used were Science Direct and PubMed, using the keywords “carotenoids”, “gut microbiota”, and “interaction”. The articles were categorized, based on their abstracts, into carotenoids in human health and the interaction of carotenoids with the gut microbiota.

**Figure 3 nutrients-15-02265-f003:**
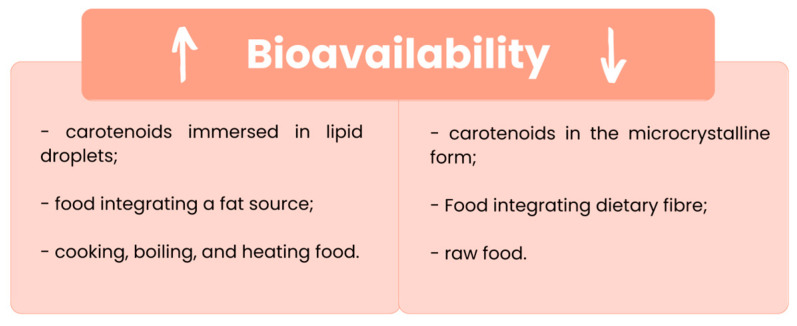
Conditioning factors that enhance (↑) or decrease (↓) the carotenoids’ bioavailability. [1,8,20,37].

**Figure 4 nutrients-15-02265-f004:**
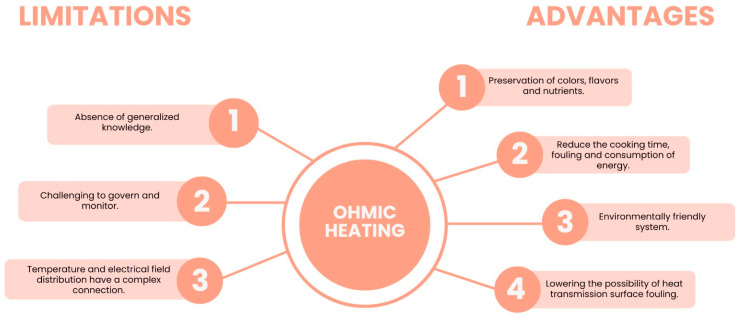
Advantages and limitations of using Ohmic heating (OH) in the extraction of bioactive compounds. Adapted from [11].

**Figure 5 nutrients-15-02265-f005:**
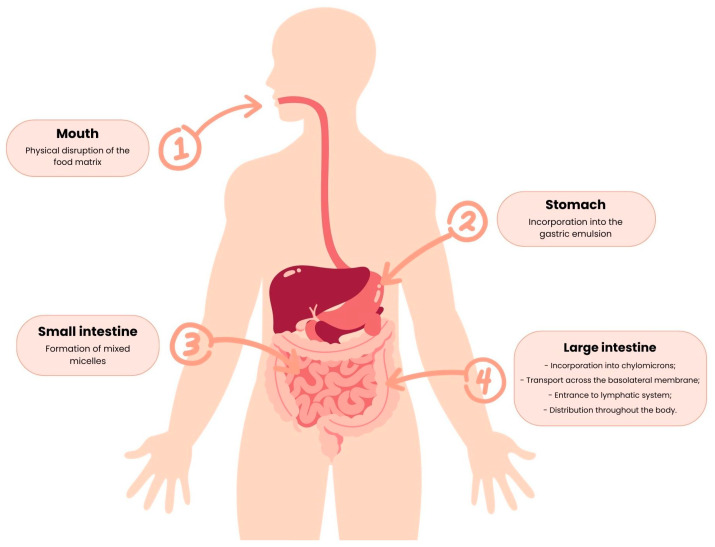
Major steps of the carotenoid absorption mechanism.

**Figure 6 nutrients-15-02265-f006:**
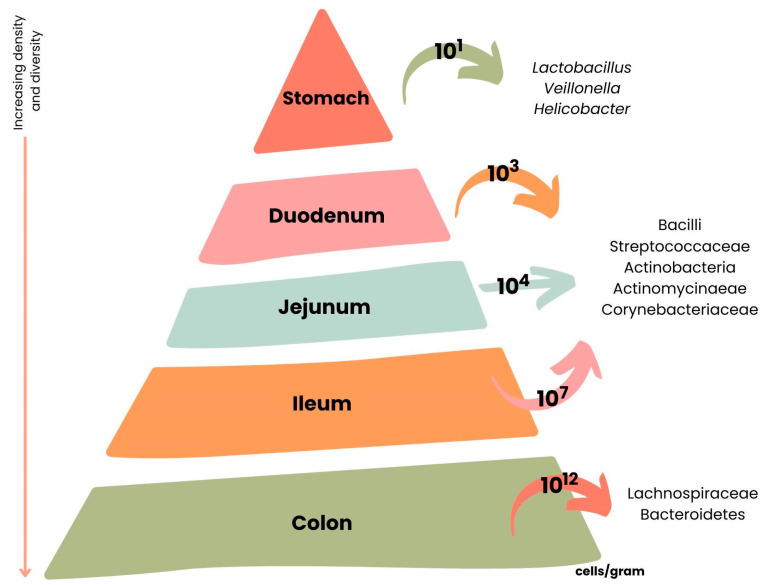
Variations across the length of the gastrointestinal tract (GIT) according to microbial cell number and composition. Adapted from [67].

**Table 1 nutrients-15-02265-t001:** Principal biological functions of β-carotene, lutein, and lycopene.

Carotenoid	Biological Functions	References
β-carotene	Stimulates the proliferation of lymphocytes; reduces the low-density lipoprotein (LDL) susceptibility to oxidation; activates cell communication; reduces inflammation; improves cardiovascular health.	[2,20,21]
Lutein	Scavenges oxygen intermediates; blue light filter; maintenance of eye health; decreases the proliferation of breast cancer cells; reduces oxidative stress and apoptosis.	[1,21,22,23,24]
Lycopene	Inhibits lipid peroxidation; eliminates reactive oxygen species (ROS); reinforces the immune system; free radical quencher; prevents skin damage.	[2,21]

**Table 2 nutrients-15-02265-t002:** Typical intestinal microbiota metabolites and their roles in health and diseases. Adapted from [71].

Groups	Typical Metabolites	Specific Function	Associated Diseases	References
Short-chain fatty acids	Acetate, propionate, butyrate, hexanoate, isovalerate, isobutyrate.	Regulation of intestinal microbiota composition, barrier integrity, and hormone production.	Diabetes, obesity, colorectal cancer, Crohn’s and Parkinson’s diseases.	[87,88,89,90,91,92,93]
Bile acids	Cholate, hyocholate, deoxycholate, glycocholate, hyodeoxycholate.	Regulation of intestinal microbiota composition, hormones, immunity, and motility.	Amyotrophic lateral sclerosis, cancer, Alzheimer’s, and Parkinson’s diseases.	[94,95,96,97,98]
Gases	H_2_S, H_2_, CO_2_, CH_2_, CH_4_, NO.	CH_4_ slows intestinal motility; H_2_S regulates intestinal inflammation and motility; NO mediates gastric mucosal protection.	Parkinson’s disease, colitis, ulcer.	[85,99,100,101,102]
Vitamins	Vitamins B2, B3, B5, B6, B9, B12, and K.	Involved in cellular metabolism, modulate immune function and cell proliferation, supply vitamins for hosts.	Vitamin-associated diseases such as schizophrenia and dementia.	[103,104]
Lipids	Conjugated fatty acids, cholesterol, lipopolysaccharides (LPS).	Conjugated fatty acids regulate the immune system; cholesterol acts as a material base for bile acid synthesis; LPS triggers systemic inflammation.	Non-alcoholic fatty liver disease, hyperinsulinemia, hypercholesterolemia.	[105,106]
Neurotransmitters	Dopamine, catecholamines, 5-HT, GABA.	Regulate intestinal motility, memory, and stress responses.	Parkinson’s disease, autism.	[85,107,108]
Choline metabolites	Dimethylglycine, methylamine, dimethylamine.	Inhibit bile acid synthesis; promote inflammation; exacerbate mitochondrial dysfunction.	Obesity, diabetes, heart failure, hypertension.	[109,110,111]
Tryptophan and indole derivatives	Indole-3-lactic acid, indole acetic acid, indole-3-acetamide, indole, serotonin.	Influence the intestinal microbial drug resistance and virulence; regulate intestinal barrier functions, hormone secretion, and motility.	Ulcerative colitis, Crohn’s, Alzheimer’s, and Parkinson’s diseases, stroke, irritable bowel syndrome.	[112,113,114,115,116]
Others	Ethanol, triphosadenine, ruminococcin A, cytolysin, microcin B17, benzoate, hippurate, cadaverine.	Regulate intestinal response, act as antibiotics to modulate intestinal microbiota composition, supply nutrients, toxic to host cells.	*C. difficile* and *H. pylori* infections, irritable bowel syndrome, ulcerative colitis.	[105,117,118,119]

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
