# Peer review of "Carotenoids Diet: Digestion, Gut Microbiota Modulation, and Inflammatory Diseases"

_nutrients, 2023, doi:10.3390/nu15102265_

Round 1
Reviewer 1 Report
Pintado and coauthors are reviewing the current understanding of the effect of the carotenoid diet on gut health and gut microbiota. This is a well-written review article pointing out the importance of the carotenoid diet and areas that need investigation for future studies. First, however, I have a few questions and suggestions.
1. In the article, when you mention that beta-carotene has to be consistent, either use the symbol for the "beta" or write "beta."
2. Line 255 - Is any specific soluble protein inhibiting the process? or Is this affected by any soluble protein?
3. Line 358 - What do you mean by the efficiency of carotenoids? Can you reword this sentence for better clarity?
4. Is there any known carotenoid processing enzymes in gut microbiota like CCDs in plants and BCOs in animals? If so, please include a few sentences about these processing enzymes that would improve the review article.
Author Response
Dear Reviewer,
First, the authors sincerely acknowledge the interest demonstrated in our work and the availability to reconsider a revised version of this manuscript.
We want to thank all your positive inputs and suggestions, which contribute to improving and enriching this manuscript.
The answers are given just after the transcription of your comments, and new information is added to the article with tracked changes as requested in the revised version.
In the article, when you mention that beta-carotene has to be consistent, either use the symbol for the "beta" or write "beta."
R. The symbol of “beta” was uniform throughout the document.
Line 255 - Is any specific soluble protein inhibiting the process? or Is this affected by any soluble protein?
R. The effect of proteins was described in the manuscript.
Line 358 - What do you mean by the efficiency of carotenoids? Can you reword this sentence for better clarity?
R. The sentence was improved accordingly.
Is there any known carotenoid processing enzymes in gut microbiota like CCDs in plants and BCOs in animals? If so, please include a few sentences about these processing enzymes that would improve the review article.
R. The enzymatic activity in the gut still needs to be clarified, but the authors included some information about these enzymes.
Reviewer 2 Report
Rocha and co-authors have made a comprehensive review on carotenoids with a focus on gut microbiota and inflammatory diseases. I have a few comments to enhance readers’ interests on reading this review.
(1) Principles and details (advantages and disadvantages) of the methods on super-critical fluid extraction (SFE), high hydrostatic pressure (HHP), and ohmic heating (OH), respectively, in section 3 on page 6.
(2) In 4.4 Absorption, can the authors elaborate the mechanisms of carotenoid absorption facilitated by transporters, e.g. SR-BI (spell out), CD36 and NPC1L1 (spell out)?
(3) It is not clear to me if secondary metabolites (what are they?) of carotenoids can be derived from gut microbiota (which microbiota and which intestinal segments?), and how they are linked to the pathogenesis of diseases in section 8.
(4) The conclusions on section 9 are too general and lack key messages.
Author Response
Dear Reviewer,
First, the authors sincerely acknowledge the interest demonstrated in our work and the availability to reconsider a revised version of this manuscript.
We want to thank all your positive inputs and suggestions, which contribute to improving and enriching this manuscript.
The answers are given just after the transcription of your comments, and new information is added to the article with tracked changes as requested in the revised version.
(1) Principles and details (advantages and disadvantages) of the methods on super-critical fluid extraction (SFE), high hydrostatic pressure (HHP), and ohmic heating (OH), respectively, in section 3 on page 6.
R. The content was improved accordingly.
(2) In 4.4 Absorption, can the authors elaborate the mechanisms of carotenoid absorption facilitated by transporters, e.g. SR-BI (spell out), CD36 and NPC1L1 (spell out)?
R. The content was improved accordingly.
(3) It is not clear to me if secondary metabolites (what are they?) of carotenoids can be derived from gut microbiota (which microbiota and which intestinal segments?), and how they are linked to the pathogenesis of diseases in section 8.
R. This content was improved accordingly.
(4) The conclusions on section 9 are too general and lack key messages.
R. The conclusion was improved accordingly.